# Effects of Vitamin D Supplementation on COVID-19 Related Outcomes: A Systematic Review and Meta-Analysis

**DOI:** 10.3390/nu14102134

**Published:** 2022-05-20

**Authors:** Banafsheh Hosseini, Asmae El Abd, Francine M. Ducharme

**Affiliations:** 1Clinical Research and Knowledge Transfer Unit on Childhood Asthma, Research Centre, Sainte-Justine University Health Centre, Montreal, QC H3T 1C5, Canada; asmae.el.abd.hsj@ssss.gouv.qc.ca (A.E.A.); francine.m.ducharme@umontreal.ca (F.M.D.); 2Department of Pediatrics, Faculty of Medicine, University of Montreal, Sainte-Justine Hospital, Montreal, QC H3C 3J7, Canada; 3Department of Social and Preventive Medicine, Public Health School, University of Montreal, Montreal, QC H3N 1X9, Canada

**Keywords:** vitamin D, COVID-19, hospitalization, mortality, ICU admission

## Abstract

The COVID-19 outbreak has rapidly expanded to a global pandemic; however, our knowledge is limited with regards to the protective factors against this infection. The aim of this systematic literature review and meta-analysis was to evaluate the impact of vitamin D supplementation on COVID-19 related outcomes. A systematic search of relevant papers published until January 2022 was conducted to identify randomized controlled trials (RCTs) and non-randomized studies of intervention (NRISs). The primary outcomes included the risk of COVID-19 infection (primary prevention studies on uninfected individuals), hospital admission (secondary prevention studies on mild COVID-19 cases), and ICU admission and mortality rate (tertiary prevention studies on hospitalized COVID-19 patients). We identified five studies (one RCT, four NRISs) on primary prevention, with five (two RCTs, three NRISs) on secondary prevention, and 13 (six RCTs, seven NRISs) on tertiary prevention. Pooled analysis showed no significant effect on the risk of COVID-19 infection. No meta-analysis was possible on hospitalization risk due to paucity of data. Vitamin D supplementation was significantly associated with a reduced risk of ICU admission (RR = 0.35, 95% CI: 0.20, 0.62) and mortality (RR = 0.46, 95% CI: 0.30, 0.70). Vitamin D supplementation had no significant impact on the risk of COVID-19 infection, whereas it showed protective effects against mortality and ICU admission in COVID-19 patients.

## 1. Introduction

The Coronavirus disease (COVID-19) outbreak has been regarded as one of the deadliest pandemics after the Spanish flu of 1918, with approximately 193 million confirmed cases and more than 6.2 million deaths as of 11 April 2022 [1]. As vitamin D strengthens innate and adaptive cellular immunity, it could reduce the survival and replication of respiratory viruses [2]. In a systematic review and individual patient meta-analysis on over 70,000 participants from 45 randomized controlled trials (RCTs), vitamin D supplementation was associated with a significantly lower risk of acute respiratory tract infection, with a trend toward greater effects in vitamin D deficient individuals and those supplemented with no bolus doses [2]. Whether vitamin D supplementation can prevent or reduce the severity of COVID-19 infection remains to be clarified.

There are conflicting data on the impact of vitamin D status on COVID-19 related outcomes, with some observational and intervention studies reporting an inverse association between vitamin D supplementation and COVID-19 mortality [3,4,5,6,7] or severity [3,8,9,10,11], while others observed no significant effects [12,13,14,15]. In a 2021 systematic review and meta-analysis of 27 observational studies, vitamin D deficiency was associated with higher disease severity and mortality, but no significant association was found with the risk of acquiring COVID-19 infection [16]. However, 74% of contributing studies were classified at high risk of bias, with no intervention studies included [16]. The latest (2022) and most comprehensive meta-analysis of 13 intervention studies (three RCTs, 10 non-randomized intervention studies (NRIS)) published before June 2021 [17] reported a significant reduction in intensive care unit (ICU) admission and mortality with vitamin D supplementation. Several additional intervention studies have been published since June 2021, which may shed more light of the role of supplemental vitamin D in COVID-19.

The objectives of this systematic review of intervention trials were to quantify the impact of vitamin D supplementation on: (1) the risk of SARS-CoV2 infection (primary prevention) in non-infected subjects; (2) the hospital admission rate (secondary prevention) in asymptomatic or mildly symptomatic infected ambulatory individuals; and (3) the rate of ICU admission or death (tertiary prevention) in hospitalized patients with COVID-19 disease.

## 2. Materials and Methods

We conducted a systematic review of intervention trials testing the impact of vitamin D supplementation as a primary, secondary or tertiary prevention against COVID-19 infection, morbidity and mortality. The study protocol has been registered in PROSPERO (registration number: CRD42021254424).

### 2.1. Search Strategy

A systematic search of relevant papers published until January 2022 was conducted using PubMed, Cochrane, CINAHL, and EMBASE. The search strategy was based on the following Medical Subject Headings search terms: (vitamin D OR cholecalciferol OR ergocalciferol OR 1,25(OH)D OR 25(OH)D, *cholecalciferol OR Calcifediol OR calcitriol OR *Dihydroxyvitamin D3 OR *D3 OR *D2) AND (COVID-19 OR SARS-CoV-2 OR coronavirus OR Coronaviridae OR Severe Acute Respiratory Distress Syndrome OR Severe acute respiratory syndrome coronavirus OR cytokine releasing syndrome OR cytokine storm). We searched clinicaltrials.gov, up until January 2022, for recently completed trials on vitamin D and COVID-19 and contacted authors to inquire about publication status. The reference lists of retrieved articles, systematic reviews, review articles, and clinical trial registration websites were also searched to identify other relevant studies.

### 2.2. Study Selection

Studies were eligible if they were randomized RCTs or NRISs, that is, quasi-experimental studies, cohorts, and case–control studies [18], testing the impact of vitamin D supplementation, with no time restriction. The following designs were excluded: animal models, cross-sectional studies, in vitro studies, systematic reviews, narrative reviews, opinion papers, case studies, and conference papers. Studies were also excluded if testing any other vitamin or mineral supplements in addition to vitamin D.

Study participants were humans of all ages, genders, or ethnicities who were either non-infected (primary prevention) or infected (secondary and tertiary prevention) with SARS-CoV-2.

The intervention was supplementation of vitamin D provided in any dose, format (oral vs. non-oral), or frequency (bolus, daily, weekly, etc.). Subjects who received vitamin D supplementation comprised the intervention arm, whereas those who received placebo or no vitamin D supplementation constituted the control arm. Standard therapy was permitted as a co-intervention, if provided to both groups.

The primary outcome for studies pertaining to primary prevention was COVID-19 incidence, defined as a positive SARS-CoV-2 infection (e.g., reverse transcription-polymerase chain reaction or serology); secondary outcomes included emergency department (ED) visits due to COVID-19, duration, and severity of the symptoms. For secondary prevention studies, the primary outcome was the rate of hospital admission due to COVID-19 infection; secondary outcomes included severity of the COVID-19 symptoms. The co-primary outcomes for tertiary prevention studies were COVID-19 mortality rate and ICU admission rate. Secondary outcomes included length of ICU admission, need for invasive ventilation, and inflammation markers. Of note, as secondary analyses, we allowed primary prevention studies to contribute to secondary and tertiary prevention outcomes, and secondary prevention studies to contribute to tertiary prevention outcomes.

Clinical or laboratory adverse health events attributed to vitamin D supplementation (such as hypercalcemia, renal lithiasis, serum 25(OH)D levels above 250 nmol/L) were sought in all studies.

### 2.3. Data Extraction and Assessment

Two authors (BH and AEA) reviewed and evaluated each citation by title and then abstract; and if potentially relevant, they performed full text assessments for study selection. Study details were extracted and recorded into a custom-designed database. Data were extracted independently by two authors (BH and AEA) and comprised the population characteristics, intervention, and outcomes (adjusted values were included in the meta-analysis, where possible). Any disagreement on study selection or data extraction was solved by consensus or input of a third reviewer (FMD). Guidelines from Preferred Reporting Items for Systematic Review and Meta-analysis (PRISMA) were followed throughout [19].

### 2.4. Risk of Bias and Quality Assessment

Eligible studies were assessed independently for their methodological quality by two reviewers (BH and AEA). For randomized clinical trials, methodological quality was assessed by the Cochrane Handbook risk of bias tool [20], based on the random sequence generation, allocation concealment, blinding of participants and personnel, blinding of outcome assessment, incomplete outcome data, selective reporting, and other sources of bias, including industry funding. Each study was rated as: low risk of bias, some concerns, or high risk of bias. Non-randomized studies of interventions were assessed based on a standardized critical appraisal checklist designed by the American Dietetic Association [21], which has been shown to have higher inter-observer agreement compared to the Cochrane risk-of-bias tool [22]. The tool comprises four relevance questions that address the applicability of the study findings to practice and 10 validity questions that address scientific rigor, including risk of bias. Based on study assessments by reviewers, each study’s risk for bias was rated as: high risk of bias (negative), low risk of bias (positive), or some concerns (neutral). Again, any discordance on methodological quality was solved by consensus or input of the third reviewer (FMD).

### 2.5. Statistical Methods

Meta-analyses were performed using the Review Manager (RevMan) computer program (Version 5.4.1, London, United Kingdom, The Cochrane Collaboration, 2020.). The risk ratio with 95% confidence interval (CI) was reported for dichotomous outcomes, with the mean difference with 95% CI reported for continuous outcomes. Adjusted values were included in the meta-analysis where reported. Appreciable heterogeneity was assumed if I^2^ > 50% and *p* < 0.1 [23]. Meta-analyses using fixed effect modeling were performed if I^2^ < 50%, otherwise a random effect modeling was used. Summary estimates (95% CI) are presented with their respective I^2^, separately for RCTs and NRISs, as well as pooled, with chi-square test reported for subgroup comparisons. Studies were stratified by type of prevention (primary, secondary and tertiary prevention). A priori stated subgroup analyses included: study design (RCT vs. NRIS), vitamin D dosing regimen (daily or weekly without bolus dosing vs. a regimen including at least one bolus of 30,000 IU or more), and baseline vitamin D status (serum 25(OH)D level <25 nmol/L (indicating deficiency) vs. ≥25 nmol/L), for each primary outcome. A sensitivity analysis was conducted on primary outcomes after excluding studies with an uncertain or high risk of bias. Tests for funnel plot asymmetry were used, provided there were at least 10 included studies in the meta-analysis as per the Cochrane’s recommendation [24].

## 3. Results

### 3.1. Search Results

The literature search identified 752 publications (747 through electronic databases and 5 via other resources including contact with researchers) (Figure 1). After excluding 9 duplicates, 743 citations were assessed based on the title and/or abstract, and 49 full-text articles were retrieved for further review. Twenty-three studies were included.

### 3.2. Study Characteristics

Nine studies were RCTs [3,8,9,10,12,13,25,26,27] and fourteen, NRIS [4,5,6,11,14,15,28,29,30,31,32,33,34,35]. All trials were performed in adults (≥18 years), with the exception of one study that included participants over 15 years old [11] (Table 1). Overall, 1548 (873 intervention: 675 control) participants were included in clinical trials and 5,868,641 (9764 intervention: 5858877 control) participants were included in NRIS. Most studies were conducted in Spain [3,6,14,25,26,33,34], with three in France [4,5,28] and Italy [15,30,35], two in India [8,13], and one each in Brazil [12], Britain [31], Canada [27], Iraq [11], Saudi Arabia [9], Turkey [29], Mexico [10], and United States [32]. A total of five studies (one RCT [27] and four NRIS [11,14,31,32]) targeted vitamin D supplementation as primary prevention, five studies (two RCTs [8,10] and three NRIS [4,15,30]), as secondary prevention and thirteen studies (six RCTs [3,9,12,13,25,26] and seven NRIS [5,6,28,29,33,34,35]), as tertiary prevention.

### 3.3. Risk of Bias

Two studies on primary prevention were assessed as low risk of bias [14,27], whereas all of those pertaining to secondary prevention were evaluated as having some concerns [4,8,10,15,30]. In the tertiary prevention group, three RCTs [3,12,26] and one NRIS [33] were rated as low risk of bias; the remainder had some quality concerns (Table 2 and Table 3).

### 3.4. Primary Prevention

One RCT [27] and four NRISs (one prospective cohort [31] and three retrospective cohort studies [11,14,32]) assessed the association between vitamin D supplementation, initiated prior to COVID-19 infection, and the risk and severity of SARS-CoV2 infection. Studies administered oral vitamin D supplementation, either as a bolus dose of 100,000 IU followed by weekly supplementation (10,000 IU) for an average of 4–6 weeks in the single RCT [27] or as daily supplementation (ranging from <1000 IU to >4000 IU) in the four NRISs [11,14,31,32]. The study duration varied from less than a week to over a year. Baseline 25(OH)D was reported in two studies (one RCT [27] and one cohort [31]); the reported average or median was close to 50 nmol/L in both studies.

In the four studies (one RCT [27] and three cohort [14,31,32]) contributing data to risk of COVID-19, vitamin D supplementation did not significantly reduce the risk of COVID-19 (N = 5,865,355, RR = 0.91, 95% CI: 0.81, 1.02, I^2^ = 0), with no significant impact of study design on the magnitude of effect (RCT: N = 33 subjects; RR = 0.28, 95% CI: 0.01, 6.43 vs. NRIS: N = 5,865,322 subjects; RR= 0.91, 95% CI: 0.82, 1.02; subgroup difference *p* > 0.05); the impact did not reach statistical significance in the RCT (Figure 2). There was also no statistically significant impact of vitamin D regimens on the observed effect (bolus: RR = 0.28 vs. non bolus: RR = 0.91, subgroup difference *p* > 0.05) (Appendix A). Subgroup analysis of the baseline vitamin D status could not be performed due to absence of trials focusing on vitamin D deficiency among those describing baseline values. No significant risk reduction following vitamin D supplementation was found in the sensitivity analysis focusing on studies at low risk of bias (n = 2 studies [14,27], combined pool RR = 0.93, 95% CI: 0.82, 1.05) (Appendix A).

Only one NRIS [11], derived from a retrospective cohort study, contributed to the secondary outcome, the number of hospital visits (vitamin D: 32.3% vs. control: 46%), and thus no meta-analysis was possible. No adverse side effect associated with vitamin D toxicity was reported in any of the studies.

### 3.5. Secondary Prevention

Two RCTs [8,10] and three NRISs (one quasi-experimental [4], one prospective cohort [30] and one retrospective cohort [15]) assessed the association between vitamin D supplementation and COVID-19 related outcomes in mildly symptomatic COVID-19-infected ambulatory subjects. Daily supplementation (10,000 IU for 14 days [10]) was tested in one RCT; oral boluses (60,000 IU/day for 7–14 days [8] or 80,000 IU every 2–3 months [4] or 50,000 IU per month [30]) were used in three studies, and daily vitamin D supplementation (mean dose >1800 IU/day) [15] in the remaining NRIS. The baseline 25(OH)D was reported in three of five studies: it was lower than 25 nmol/L in both groups in one RCT [8], whereas the median varied between 50.5 and 58.5 nmol/L in one RCT [10] and between 30 and 80 nmol/L in the cohort study [15].

Only one study, a retrospective cohort [15], reported our primary outcome; no statistically significant impact of vitamin supplementation was noted on hospital admission (adjusted OR: 1.30, 95% CI; (0.51, 3.32)). As for secondary outcomes, in two (one RCT and one quasi-experimental) studies, the supplemented group presented less severe COVID-19 symptoms compared with controls [4,10] with symptoms measured differently, preventing aggregation; these were defined by the presence of fewer than three symptoms at day 14 in the RCT [10] or having a lower COVID-19 ordinal scale for clinical improvement (OSCI) score in the quasi-experimental study [4]. No adverse health event attributable to vitamin D toxicity was reported in any of the studies.

### 3.6. Tertiary Prevention

Six RCTs [3,9,12,13,25,26] and seven NRISs (two quasi-experimental studies [5,28], two prospective cohorts [29,33], two retrospective cohorts [6,35] and one retrospective case–control design [34]) evaluated the impact of vitamin D supplementation in patients hospitalized with COVID-19.

In five studies (one RCT [12], four NRISs [5,29]), the intervention included one or more oral (or intramuscular [29]) boluses of vitamin, varying from a bolus of 200,000 IU [12], 80,000 IU [5], and 300,000 IU [29], or two doses of 200,000 IU for 2 successive dates [35] or 60,000 IU/day for 8–10 days [13]. In the remaining six studies (four RCTs [3,9,25,26], two NRISs [6,33]), non-bolus supplementation was provided daily or weekly, varying from 2000 IU/day for 6 weeks [26] to 10,810 IU 1–2 times per week for at least 4 weeks [3,6,25,33], with one RCT using 5000 IU daily for 2 weeks [9]. Additionally, in two NRISs [28,34], the intervention group was classified as being supplemented with vitamin D ranging from 10,000 IU/month to 50,000 IU/month for 3 months. Of note, vitamin D supplementation was initiated after diagnosis of COVID-19 in all but two studies, in which it was started more than 3 months prior to hospital admission [28,34].

The baseline 25(OH)D was reported in all but one RCT [3]; the mean baseline value was consistently above 25 nmol/L in both intervention and control groups (mean or median ranging from 30 to 63.5 nmol/L). Five of the seven NRISs reported baseline 25(OH)D [28,29,33,34,35], with two studies reporting a mean or median value of less than 25 nmol/L in the intervention group only [35] or in both groups [29]; the mean or median reported in the remaining studies ranged from 30 to 73 nmol/L.

Eleven tertiary prevention studies (four RCTs [3,12,13,25] and seven NRISs [5,6,28,29,33,34,35]) contributed data to the main outcome. Compared to controls, an overall statistically significant reduction in the risk of COVID-19 mortality was observed (n = 3391; RR = 0.52, 95% CI: 0.36, 0.75, I^2^ = 54%), with no significant subgroup difference or heterogeneity between study designs (RCT: n = 1330; 0.56 (0.25, 1.25) vs. NRIS: n = 2061; 0.56 (0.32, 0.72); Chi^2^ = 0.04, *p* = 0.84, I^2^ = 0%); however, the impact did not reach statistical significance in the RCT subgroup (Figure 3a).

A funnel plot of the effect size of vitamin D supplementation on COVID-19 mortality showed some degree of asymmetry (Appendix A). Sensitivity analysis focusing on the three (two RCTs [3,12]; one NRIS [33]) studies at low risk of bias showed an effect size of similar magnitude on COVID-19 mortality that did not reach statistical significance (N = 1151 subjects, pooled RR = 0.48, 95% CI: 0.12, 1.87) (Appendix A).

Subgroup analysis of tertiary prevention studies showed a significant group difference in the magnitude of effect by vitamin D dosing regimen (bolus: RR = 0.69, vs. no bolus: RR = 0.37; subgroup difference *p* = 0.04) (Appendix A). Protective effects of vitamin D supplementation remained significant only in studies with baseline 25(OH)D level ≥ 25 nmol/L (25(OH)D ≥ 25 nmol/L: RR = 0.50 vs. 25(OH)D < 25 nmol/L: RR = 0.92 vs. 25(OH)D not specified: RR = 0.31); subgroup difference *p* > 0.05) (Appendix A).

As a secondary analysis, adding the data of three NRISs targeting secondary prevention [4,15,30] contributing to this co-primary outcome, showed again a similar reduction in the mortality rate in the vitamin D supplemented group (N = 3725, RR = 0.46, 95% CI: 0.30, 0.70, I^2^ = 58%), with no significant subgroup differences in study design (RCT: RR = 0.52 vs. NRIS: RR = 0.44; subgroup difference *p* > 0.05, I^2^ = 0%; Figure 3b). Subgroup analysis of all studies based on vitamin D regimens showed no significant group difference in the magnitude of effect by vitamin D dosing regimen (bolus: RR = 0.57 vs. no bolus: RR = 0.50; subgroup difference *p* > 0.05; Appendix A). The preventive effects of vitamin D supplementation remained significant in studies with baseline 25(OH)D level ≥ 25 nmol/L; however, no statistically significant impact was observed in those with baseline vitamin D deficiency in at least one group (25(OH)D ≥ 25 nmol/L: RR = 0.63 vs. 25(OH)D < 25 nmol/L: RR = 0.92 vs. 25(OH)D not specified: RR = 0.31; subgroup difference *p* = 0.04; Appendix A).

COVID-related ICU admission was reported in four RCTs [3,12,13,25] and three NRISs [33,34,35]; statistically significantly lower risks of ICU admission in the vitamin D supplementation group were observed in RCTs (RR = 0.36, 95% CI: 0.14, 0.89, I^2^ = 82%), NRISs (RR = 0.32, 95% CI: 0.14, 0.75, I^2^ = 68), and pooling all designs (RR = 0.35, 95% CI: 0.20, 0.62, I^2^ = 75%) with no significant group difference between designs (*p* = 0.86, I^2^ = 0%) (Figure 4).

The preventive effects of vitamin D therapy on ICU admission rate remained significant in the sensitivity analysis focusing on studies at low risk of bias (n = 3 studies [3,12,33], combined pool RR = 0.25, 95%CI: 0.07, 0.86) (Appendix A).

Subgroup analysis of studies showed a significant group difference in the magnitude of effect by vitamin D dosing regimen (bolus: RR = 0.71 vs. no bolus: RR = 0.22), with a significant subgroup difference (*p* < 0.0001) (Appendix A). Moreover, protective effects of vitamin D supplementation on COVID-19 ICU admission rate remained significant in studies with baseline 25(OH)D level ≥25 nmol/L as well as those that did not report baseline serum values (25(OH)D ≥ 25 nmol/L: RR = 0.37 vs. 25(OH)D < 25 nmol/L: RR = 0.61, vs. 25(OH)D not specified: RR = 0.35), with a significant subgroup difference (*p* = 0.04) (Appendix A).

As for secondary outcomes, three studies (two RCTs [12,13] and one prospective cohort [29]) assessed the impact on the need for invasive mechanical ventilation, showing a 50% (27%, 93%) reduced rate of invasive ventilation in the supplemented group with similar effect size in RCTs and NRISs with no significant group difference (RCT: RR = 0.58 vs. NRIS: RR = 0.24, I^2^ = 0%, subgroup difference *p* > 0.05, low quality) (Appendix A).

Two RCTs [12,13] and one retrospective case–control study [34] evaluated the impact of supplementation on length of hospital stays in patients with COVID-19 infection; no significant impact was observed (Appendix A).

In the three RCTs [9,13,26] evaluating the effects of supplementation on inflammation, two studies [9,26] showed no impact, while one study [13] reported a significant reduction in serum levels of c-reactive protein (CRP) following vitamin D supplementation (difference (pre-post) vitamin D: 51 (10, 113) mg/L vs. non-vitamin D: 5 (−3, 39) mg/L).

No severe adverse events associated with vitamin D toxicity were reported in any of the studies, with the exception of one RCT [12] that reported that one patient vomited after vitamin D administration.

## 4. Discussion

The present systematic review and meta-analysis revealed that while vitamin D supplementation did not significantly reduce the risk of COVID-19 infection, it was associated with a lower mortality rate and risk of ICU admission in patients hospitalized with COVID-19.

In the meta-analysis of four primary prevention studies, no statistically significant association between vitamin D supplementation and risk of SARS-CoV2 infection was observed, although a 10% reduction approaching statistical significance was noted. The type of vitamin D regimen (bolus vs. no bolus) did not significantly affect the result. Whether the magnitude of effect varied based on baseline 25(OH)D serum level could not be examined as no studies focusing on subjects with vitamin D deficiency were reported. As for the studies that targeted secondary prevention, no meta-analysis was possible on the effects of vitamin D supplementation on hospital admission or symptom severity due to unavailability of data. Furthermore, vitamin D supplementation, usually administered as bolus doses in patients infected with COVID-19, showed a significant reduction by half in COVID-19 mortality rate and by 65% in ICU admission rate. Of importance, the magnitude of benefit was not significantly different in the RCTs and NRISs. Moreover, our sensitivity analyses limited to studies with a low risk of bias did not change the direction of any pooled effect; however, with a lower power, the preventive effects remained statistically significant only for ICU admission rate.

Our null finding on the association between vitamin D supplementation and risk of SARS-CoV2 infection is in concordance with a Mendelian randomization study that used genetic variants strongly associated with higher vitamin D levels and that showed no association between vitamin D status and the likelihood of being infected with COVID-19 [36]. Collectively, the findings point toward no or a small reduction in the risk of SARS-CoV2 infection associated with vitamin D supplementation, in adult patients who are not vitamin D deficient at baseline; however, given the large confidence interval and the paucity of adequately powered RCTs, there were insufficient data to allow a firm conclusion. To our knowledge, no previous meta-analysis explored the effects of vitamin D supplementation in mild cases of COVID-19. A previous study that used Mendelian randomization to assess the effect of increased vitamin D on COVID-19 outcomes also showed no significant difference in risk of hospitalization and did not support the use of vitamin D supplementation in the general population to prevent COVID-19 outcomes [36]. A few previous systematic literature reviews and meta-analyses have addressed the association between vitamin D status and COVID-19 related mortality and ICU admission rate and showed conflicting results [17,37,38,39]. With ten additional studies, our findings are in line with the 2021 meta-analysis of COVID-19 patients including 13 studies (10 observational with 3 RCTs), which concluded that vitamin D supplementation was significantly associated with a lower risk of ICU admission and mortality, with an effect size comparable to what we found in the present study [17]. Of note, in the 2021 meta-analysis, ICU admission rate and mortality rate were pooled together as one single outcome, whereas in this review, results for each outcome are presented separately. Moreover, our sensitivity analyses, limited to three studies with a low risk of bias, did not change the direction of any pooled effect; however, the preventive effects remained statistically significant in ICU admission rate only.

In our review, regimens without bolus doses appeared to have stronger preventive effects against both COVID-19 mortality and ICU admission rate compared to bolus doses. This observed greater benefit of supplementation without vs. with a bolus is in line with a previous meta-analysis that concluded bolus dose of vitamin D might be less effective for prevention of acute respiratory tract infection [2]. One explanation for this can be due to the wide fluctuations in serum 25(OH)D levels after using bolus doses, which could dysregulate the activity of enzymes involved in the synthesis and degradation of the active vitamin D metabolites. Such an effect can reduce the concentrations of vitamin D in extra-renal tissue and thereby attenuate the immunomodulatory effects of vitamin D supplementation [2]. We also showed that the protective effects of vitamin D against both COVID-19 mortality and ICU admission appear stronger when baseline vitamin D status is ≥25 nmol/L compared to vitamin D deficiency (<25 nmol/L). Results from further RCTs may clarify whether the threshold for beneficial effects from vitamin D supplementation varies between subjects with different baseline vitamin D status.

With regards to the effects of vitamin D supplementation on inflammatory biomarkers in COVID-19 infection, of three studies [9,13,26] evaluating the effects of supplementation on inflammation, only one [13] showed a significant reduction in serum CRP levels in the vitamin D supplemented group. Vitamin D supplementation could suppress the nuclear factor kappa B (NF-κB) pathway, which in turns may reduce systemic inflammation and production of CRP [40]. A previous meta-analysis of 10 trials involving a total of 924 participants indicated that daily vitamin D supplementation (ranging from 400 to 7143 IU for 8 to 48 weeks) significantly decreased the circulating CRP level [40]. Additional well-designed RCTs are warranted to further investigate the role of vitamin D supplementation in systemic inflammation in patients with acute respiratory infections.

To the best of our knowledge, the present study provides the most comprehensive pooled data on the effects of vitamin D supplementation on COVID-19 related outcomes. As of April 11, 2022, there were 40 studies registered at Clinicaltrials.gov on the effects of vitamin D supplementation on COVID-19 outcomes, with the majority either completed (n = 14) or still recruiting (n = 14). The majority of upcoming studies will examine the effects of vitamin D supplementation on the severity of COVID-19 (secondary and tertiary prevention), although four studies will be focused on primary prevention. Inclusion of the data from these ongoing studies in future meta-analyses will allow a clearer conclusion on the effects of vitamin D supplementation in preventing/managing COVID-19.

The strengths of this review include the broad systematic literature search, clearly defined inclusion and exclusion criteria, independent study selection, data extraction, methodological quality assessment and rigorous meta-analyses. We acknowledged several limitations. Studies varied in terms of study design, participants (baseline 25(OH)D, severity of COVID-19), and intervention (dose, regimens, duration). Although we addressed this by conducting several a priori specified subgroup analyses to shed some light on the impact of baseline 25(OH)D and dosing regimens on the magnitude of effect, we cannot rule out the possibility of confounding or effect modification of intervention. Therefore, we advise caution in the interpretation of subgroup analyses because incomplete reporting of characteristics, heterogeneity of characteristics within trials, and absence of individual patient data prevented us from conducting meta-regressions that could have better untangled the concurrent impact of study design, participant, or intervention on effect size. While over 60% of included studies used quasi-experimental or cohort designs, our analysis showed that the magnitude of effect was consistent across study designs and when restricted to studies at low risk of bias. As included studies pertained to participants over the age of 15 years old, our findings could not be generalized to children and younger adolescents.

## 5. Conclusions

Our findings suggest that vitamin D supplementation, administered in hospitalized COVID-19 patients, is associated with a significant reduction in mortality, ICU admission, and need for mechanical ventilation. Nevertheless, uncertainty remains on the characteristics of individuals with stronger protective effects and the type of intervention (e.g., dose, regimen, duration) yielding the greatest benefit. There was insufficient evidence to determine whether vitamin D supplementation can significantly decrease the risk of acquiring COVID-19 infection when taken as a primary prevention or reduce the severity and risk of hospital admission when taken as secondary prevention in asymptomatic or mildly symptomatic COVID-19 cases. An updated meta-analysis upon completion of ongoing trials is needed to expand our understanding of the effects of vitamin D supplementation on preventing and managing COVID-19.

## Figures and Tables

**Figure 1 nutrients-14-02134-f001:**
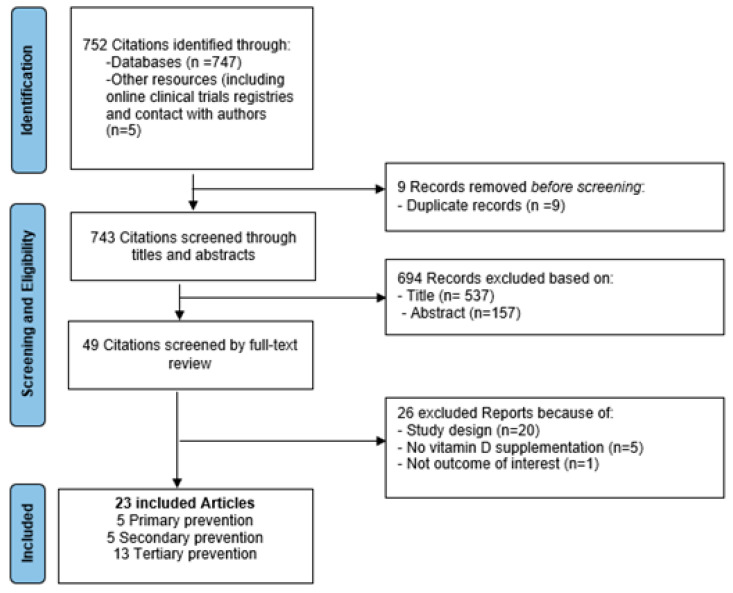
Selection process for eligible studies from all identified citations.

**Figure 2 nutrients-14-02134-f002:**
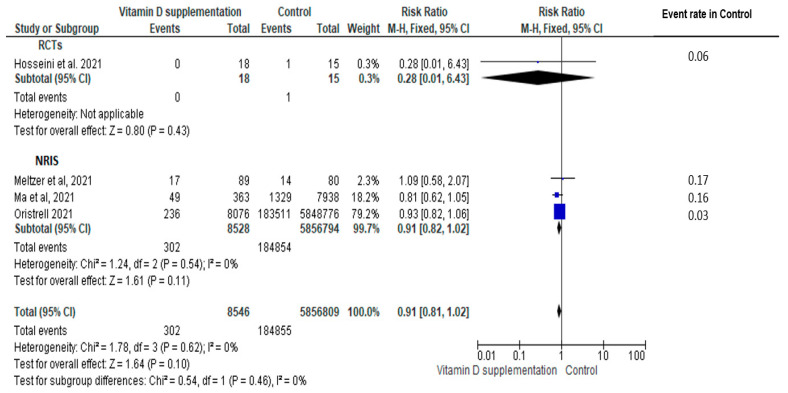
Pooled relative risk of subjects experiencing a confirmed COVID-19 infection comparing vitamin D supplementation with controls (placebo or no vitamin D supplementation). Studies are stratified by study design (randomized controlled trials (RCTs) [27] vs. non-randomized intervention studies (NRISs) [14,31,32]. Subgroup and pooled summary estimates are reported with 95% confidence intervals, analyzed with the Mantel–Haenszel fixed-effects model method. Heterogeneity was quantified by I^2^. The chi-square test served to examine group differences between study designs.

**Figure 3 nutrients-14-02134-f003:**
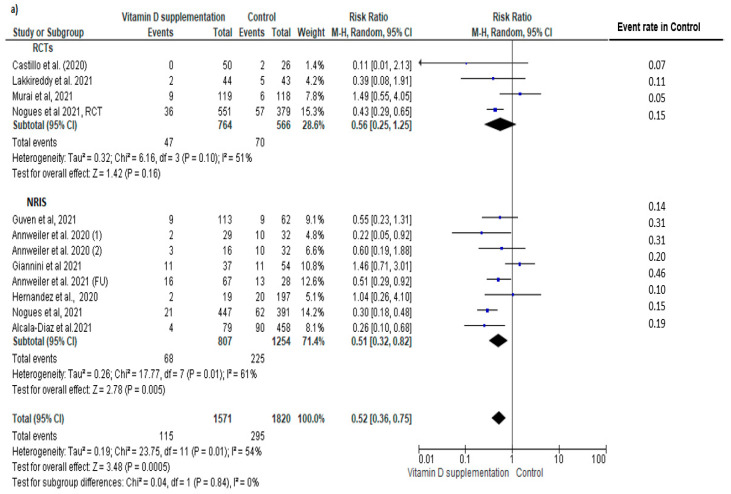
Pooled relative risk of mortality in patients with COVID-19 comparing vitamin D supplementation with controls (placebo or no vitamin D supplementation) in (**a**) tertiary prevention studies and (**b**) both secondary and tertiary prevention studies. Studies are stratified by study design (randomized controlled trials (RCTs) [3,12,13,25] vs. non-randomized intervention studies (NRIS) [4,5,6,15,28,29,30,33,34,35]. Subgroup and pooled summary estimates are reported with 95% confidence interval, analyzed with the Mantel–Haenszel fixed-effects model method. Heterogeneity was quantified by I^2^. The chi-square test served to examine group difference between study designs.

**Figure 4 nutrients-14-02134-f004:**
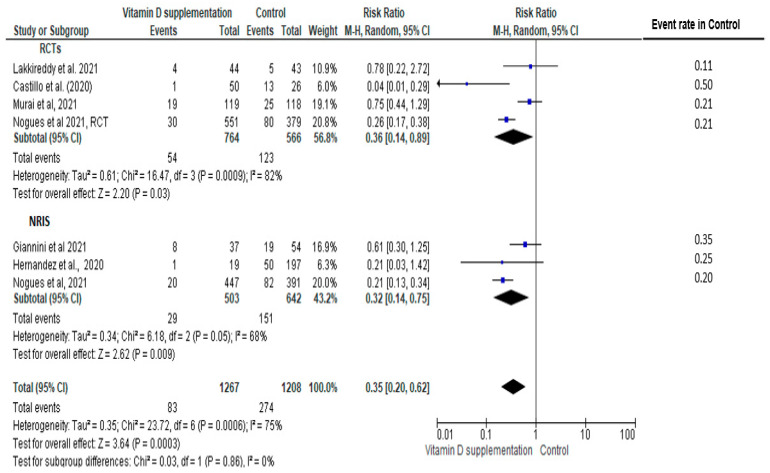
Pooled relative risk of ICU admission in patients with COVID-19 comparing vitamin D supplementation with controls (placebo or no vitamin D supplementation). Studies are stratified by study design (randomized controlled trials (RCTs) [3,12,13,25] vs. non-randomized intervention studies (NRISs) [33,34,35]. Subgroup and pooled summary estimates are reported with 95% confidence interval, analyzed with the Mantel–Haenszel fixed-effects model method. Heterogeneity was quantified by I^2^. The chi-square test served to examine group difference between study designs.

**Table 1 nutrients-14-02134-t001:** Characteristics of studies on the effects of vitamin D supplementation and COVID-19 related outcomes.

Reference	Design, Setting	Participants	Duration of Intervention	Treatment Arms	Baseline Serum 25OHD (nmol/L)	
					Vitamin D	Control
**Primary Prevention**						
Hosseini et al. [27], 2021	RCT, Canada	Unvaccinated healthcare workers (25–58 years old, male: 5.9%)	4–10 weeks	**Intervention:** vitamin D: bolus 100,000 IU + 10,000 IU/week **(n = 19)**; **Control:** placebo **(n = 15)**	49.56 ± 26.64	48.02 ± 15.16
Abdulateef et al. [11], 2021	Retrospective cohort, Iraq	Patients with COVID-19 (15–80 years old, male: 44.4%)	Not specified	**Intervention:** regularly supplemented with vitamin D prior to COVID-19 exposure **(n = 127)**, “ranging from <1000 IU/day to >4000 IU/day for <1 week to >2 weeks”; **Control:** no vitamin D supplements **(n = 300)**	Not reported	Not reported
Ma et al. [31], 2021	Prospective cohort, United Kingdom	Adults who have records of COVID-19 test results from UK Biobank (37–73 years old, male: 44.4%)	Not specified	**Intervention:** regularly supplemented with vitamin D (not specified) prior to COVID-19 exposure **(n = 363)**; **Control:** no vitamin D supplements **(n = 7934)**	56 ± 20.8	47 ± 21.1
Meltzer et al. [32], 2020	Retrospective cohort, United States	489 patients with data for a vitamin D level within 1 year before COVID-19 testing (49.2 ± 18.4 years old, male: 25.0%)	Not specified	**Intervention:** “regularly supplemented with vitamin D over the past year excluding the 14 days before testing: (≤1000 IU, 2000 IU, ≥3000 IU)” **(n = 277)**; **Control:** no vitamin D supplements **(n = 212)**	Not reported	Not reported
Oristrell et al. [14], 2021	Retrospective cohort, Spain	Patients with chronic kidney disease (70.2 ± 15.6 years old, male: 42.5%)	10 months	**Intervention:** supplemented with vitamin D prior to COVID-19 exposure (10,596 IU/day) from 1 April 2019 to 28 February 2020 **(n = 8076)**; **Control:** no vitamin D supplements **(n = 5,848,776)**	Not reported	Not reported
**Secondary Prevention**						
Rastogi et al. [8], 2020	RCT, India	Asymptomatic or mildly symptomatic cases of COVID-19 (36 to 51 years old, male: 45.0%)	7 days or more if needed	**Intervention:** vitamin D: 60,000 IU/day; **(n = 16)** (with therapeutic target 25 OHD > 125 nmol/day); **Control:** identical placebo **(n = 24)**	21.5 (17.7, 32.7)	23.8 (20.5, 31.2)
Sánchez-Zuno et al. [10], 2021	RCT, Mexico	COVID-19 outpatients 20–74 years old, male: 47.7%)	14 days	**Intervention:** 10,000 IU of vitamin D3/day **(n = 22)**; **Control:** placebo **(n = 20)**	50.5 (30.5, 114.7)	58.5 (30.25, 114)
Annweiler et al. [4], 2021	Quasi-experimental with retrospective collection of data, France	Elderly nursing-home residents infected with COVID-19 (63–103 years old, male: 23.7%)	Single bolus	**Intervention:** single oral dose of 80,000 IU vitamin D3 during COVID-19 or in the preceding month **(n = 57)**; **Control:** no vitamin D supplements **(n = 9)**	Not reported	Not reported
Cangiano et al. [30], 2021	Prospective cohort, Italy	157 residents of a nursing home after Sars-CoV-2 spread (80–100 years old, male: 28.5%)	2 months	**Intervention:** vit D supplementation: 50,000 IU/month **(n = 20)**; **Control:** no vitamin D supplementation **(n = 78)**	Not reported	Not reported
Cereda et al. [15], 2020	Retrospective cohort, Italy	COVID-19 outpatients (68.8± 10.6 years old, male: 48.4%)	3 months	**Intervention:** supplemented (“mean intake of >1800 IU/day”) **(n= 38)**; **Control:** no vitamin D supplementation **(n = 286)**	82.2 ± 37	28.2 ± 21.5
**Tertiary Prevention**						
Caballero-García et al. [26], 2021	RCT, Spain	Patients in the recovery phase post hospitalization with COVID-19 infection (62.5 ± 1.5 years old, male: 100.0%)	6 weeks	**Intervention:** vitamin D: , IU/day **(n = 15)**; **Control:** placebo **(n = 15)**	52.2 ± 4.5	53.0 ± 3.5
Castillo et al. [3], 2020	RCT, Spain	Patients hospitalized with COVID-19 infection (53.14 ± 1 0.77 years old, male: 59.0%)	4 weeks	**Intervention:** 21,280 IU/day vitamin D on day 1, 3 and 7, and then weekly until discharge or ICU admission **(n = 50)**; **Control:** no vitamin D supplementation **(n = 26)**	Not reported	Not reported
Lakkireddy et al. [13], 2021	RCT, India	Patients hospitalized with COVID-19 infection (20–83 years old, male: 75%)	8–10 days	**Intervention:** 60,000 IU/day vitamin D **(n = 44)**; **Control:** no vitamin D supplementation **(n = 43)**	40 ±15	42.5 ± 15
Murai et al. [12], 2021	RCT, Brazil	Patients hospitalized with COVID-19 infection (56.2 ± 14.4 years old, male: 46.1%)	20 days	**Intervention:** single bolus of 200,000 IU vitamin D **(n = 120)**; **Control:** placebo **(n = 120)**	53 ± 25.2	51.5 ± 20.2
Nogues et al. [25], 2021	RCT, Spain	Patients hospitalized with COVID-19 infection (30–80 years old, male: 56.0%)	30 days	**Intervention:** vitamin D: 21,620 IU on day 1, 10,810 IU on day 3, 7, 15, and 30) **(n = 551)**; **Control:** placebo **(n = 379)**	37.5 (22.5, 70)	30 (8,47.5)
Sabico et al. [9], 2021	RCT, Saudi Arabia	Patients hospitalized with COVID-19 infection (20–75 years, male: 47.8%)	2 weeks	**Intervention:** vitamin D: 5000 IU/day **(n = 36)**; **Control:** 1000 IU/day **(n = 33)**	53.4 ± 2.9	63.5 ± 3.4
Alcala-Diaz et al. [6], 2021	Retrospective cohort, Spain	Patients hospitalized with COVID-19 infection (69 ± 15 years old, male: 59.0%)	28 days	**Intervention:** vitamin D: 21,620 IU on day 1, 10,810 IU on day 3, 7, 14, 21, and 28) **(n = 79)**; **Control:** no vitamin D supplementation **(n = 458)**	Not reported	Not reported
Annweiler et al. [5], 2020	Quasi-experimental with retrospective collection of data, France	Patients hospitalized with COVID-19 infection (78- 100 years old, male: 51.0%)	Not specified	**Group 1:** regularly supplemented with vitamin D (50,000 IU/month) **(n = 29)**; **Group 2:** vitamin D supplementation initiated after COVID-19 diagnosis (80,000 IU bolus) **(n = 16)**; **Group 3:** no vitamin D supplementation **(n = 32)**	Not reported	Not reported
Annweiler et al. [28], 2021	Quasi-experimental with retrospective collection of data, France	Patients hospitalized with COVID-19 infection (78–100 years old, male: 51.0%)	Not specified	**Intervention:** regularly supplemented with vitamin D (50,000 IU/month or 800 IU/day) **(n = 67)**; **Control:** no vitamin D supplementation **(n = 28)**	61.6 ± 35.4	73.9 ± 32.1
Giannini et al. [35], 2021	Retrospective cohort, Italy	Patients hospitalized with COVID-19 infection (74.0 ± 13.0 years old, male: 75%)	2 days	**Intervention:** oral bolus of 200,000 IU vitamin D on the second and third day of hospital stay **(n = 36)**; **Control:** no vitamin D supplementation **(n = 55)**	24 (12, 42)	36 (19, 77)
Guven et al. [29], 2021	Prospective cohort, Turkey	Patients hospitalized with COVID-19 infection (74 (61–82)) years old, male: 61.0%)	Single bolus	**Intervention:** single dose of 300,000 IU intramuscularly **(n = 113)**; **Control:** no vitamin D supplementation **(n = 62)**	16.6 (12.6, 22.7)	17.8 (14.2, 20.5)
Hernandez et al. [34], 2021	Case–control, Spain	Patients hospitalized with COVID-19 infection (60.0 (59.0–75.0)) years old, male: 60.1%)	More than 3 months prior to hospital admission	**Intervention:** supplemented with vitamin D (“range from 10,000 IU/month, to 5600 IU/week or 25,000 IU/month”) **(n = 19)**; **Control:** no vitamin D supplementation **(n = 197)**	52.7 ± 14.7	34.5 ± 18
Nogues et al. [33], 2021	Prospective cohort, Spain	Patients hospitalized with COVID-19 infection (61.81 ± 15.5 years old, male: 59.0%)	28 days	**Intervention:** vitamin D: 21,620 IU on day 1, 10,810 IU on day 3, 7, 14, 21, and 28) **(n = 447)**; **Control:** no supplementation **(n = 391)**	32.5 (20.0, 60.0)	30 (20, 47.5)

Values are reported as mean ± standard deviation or median (interquartile range). Abbreviations: 25OHD, 25-hydroxyvitamin D; IU, international unit; Bold n, number.

**Table 2 nutrients-14-02134-t002:** Risk of bias summary based on Cochrane Systematic Review Guidelines for included randomized controlled trials.

Study	Selection Bias ^1^	Selection Bias ^2^	Performance Bias ^3^	Attrition Bias ^4^	Detection Bias ^5^	Reporting Bias ^6^	Overall Risk of Bias
**Primary prevention**							
**Hosseini et al.** [27]	Low	Low	Low	Low	Low	Low	Low
**Secondary prevention**							
**Rastogi et al**. [8]	Some concerns	Some concerns	Some concerns	Low	Some concerns	Some concerns	Some concerns
**Sanchez et al.** [10]	Some concerns	Some concerns	Some concerns	Low	Low	Low	Some concerns
**Tertiary prevention**							
**Castillo et al.** [3]	Low	Low	Low	Some concerns	Low	Low	Low
**Caballero-Garcia et al.** [26]	Low	Low	Low	Low	Low	Low	Low
**Lakkireddy et al.** [13]	Some concerns	Some concerns	Some concerns	Low	Low	Low	Some concerns
**Murai et al.** [12]	Low	Low	Low	Low	Low	Low	Low
**Nogues et al.** [25]	Some concerns	Some concerns	Some concerns	Low	Low	Low	Some concerns
**Sabico et al.** [9]	Some concerns	Some concerns	Some concerns	Low	Low	Low	Some concerns

^1^ Random sequence generation, ^2^ Allocation concealment, ^3^ Blinding of participants and personnel, ^4^ Incomplete outcome data, ^5^ Blinding of outcome assessment, ^6^ Selective reporting

**Table 3 nutrients-14-02134-t003:** (Bias assessment of each non-randomized intervention study based on standardized critical appraisal checklist designed by the American Dietetic Association.

Study	Q1 ^1^	Q2 ^2^	Q3 ^3^	Q4 ^4^	Q5 ^5^	Q6 ^6^	Q7 ^7^	Q8 ^8^	Q9 ^9^	Q10 ^10^	Overall Risk of Bias
**Primary prevention**											
Abdulateef et al. [11]	Yes	No	Yes	Yes	No	No	Yes	No	Yes	Yes	Some concerns
Ma et al. [31]	Yes	No	Yes	Yes	No	No	Yes	Yes	Yes	Yes	Some concerns
Meltzer et al. [32]	Yes	No	Yes	Yes	No	No	Yes	Yes	Yes	Yes	Some concerns
Oristrell et al. [14]	Yes	No	Yes	Yes	No	Yes	Yes	Yes	Yes	Yes	Low
**Secondary prevention**											
Annweiler et al. [4]	Yes	No	No	Yes	Unclear	Yes	Yes	Yes	Yes	Yes	Some concerns
Cangiano et al. [30]	Yes	No	No	No	Unclear	Yes	Yes	Yes	Yes	Yes	Some concerns
Cereda et al. [15]	No	No	Yes	Yes	No	Yes	Yes	Yes	Yes	Yes	Some concerns
**Tertiary prevention**											
Annweiler et al. [5]	Yes	No	No	Yes	Unclear	No	Yes	Yes	Yes	Yes	Some concerns
Annweiler et al. [28]	Yes	No	No	Yes	Unclear	No	Yes	Yes	Yes	Yes	Some concerns
Alcala-Diaz et al. [6]	Yes	No	No	Yes	No	Yes	Yes	Yes	Yes	Yes	Some concerns
Giannini et al. [35]	Yes	No	No	Yes	Unclear	Yes	Yes	Yes	Yes	Yes	Some concerns
Guven et al. [29]	Yes	No	Yes	Yes	Unclear	Yes	Yes	No	Yes	Yes	Some concerns
Nogues et al. [33]	Yes	No	Yes	Yes	Unclear	Yes	Yes	Yes	Yes	Yes	Low
Hernandez et al. [34]	Yes	No	No	No	Unclear	Yes	Yes	Yes	Yes	Yes	Some concerns

^1^ Q1 = Question 1: Was the research question clearly stated? ^2^ Q2 = Question 2: Was the selection of study subjects/patients free from bias? ^3^ Q3 = Question 3: Were study groups comparable? ^4^ Q4 = Question 4: Was the method of handling withdrawals described? ^5^ Q5 = Question 5: Was blinding used to prevent introduction of bias? ^6^ Q6 = Question 6: Were intervention/therapeutic regimens/exposure factors or procedures and any comparison(s) described in detail? Were intervening factors described? ^7^ Q7 = Question 7: Were outcomes clearly defined and the measurements valid and reliable? ^8^ Q8 = Question 8: Was the statistical analysis appropriate for the study design and type of outcome indicators? ^9^ Q9 = Question 9: Were conclusions supported by results with biases and limitations taken into consideration? ^10^ Q10 = Question 10: Was bias due to study’s funding or sponsorship unlikely?

## Data Availability

The datasets used and/or analyzed during the current study are available from the corresponding author on reasonable request.

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
