# Peer review of "Effects of Vitamin D Supplementation on COVID-19 Related Outcomes: A Systematic Review and Meta-Analysis"

_nutrients, 2022, doi:10.3390/nu14102134_

Round 1
Reviewer 1 Report
The aim of the present article was to review the impact of vitamin D supplementation in COVID-19 related outcomes. This is an issue of general interest. However, serious methodological flaws limit the value of the presented findings and require revision.
2.2. Study selection: Please explain what is meant by ‘‘controlled before and after studies’’ and ‘‘with no time of language restriction’’.
2.2. Study selection: ‘‘The intervention was supplementation of vitamin D provided in any dose, format (oral vs. non oral), or frequency (bolus, daily, weekly, etc.).’’ Although I completely agree that every regimen should be assessed, you should never combine clinically heterogeneous results in a meta-analysis. Please discuss clinically heterogeneous studies in the context of the systematic review and refrain from providing pooled estimates. It would also be appropriate to look for potential trends (dosage associations, i.e., better outcomes following administration of larger doses, different responses between younger and older adults or better outcomes with longer treatment duration) that may provide additional insight.
Humans of all age, gender, or ethnicity, with any health status were also included. Again, it is not acceptable to combine studies with high heterogeneity. A study involving frail older adults should not be pooled with a study on healthy young individuals.
‘‘Preferred reporting items for systematic review and meta-analysis 114 (PRISMA) was followed during data extraction and study synthesis’’. The PRISMA checklist should be abided by throughout the whole article and not in two sections.
2.4. Risk of bias and Quality assessment: Your reference [22] implies that the Quality Criteria Checklist (QCC) of the American Dietetic Association are used for the assessment of the RoB of RCTs. You mention that you implemented it in the appraisal of the RoB of non-RCTs. Is there a validated version of the QCC for cohort and case control studies? If yes, please provide it. Did you use this version of the QCC?
The evaluation of the strength of evidence using the GRADE-PRO approach should be only applied by experts. The RoB assessment is of pivotal importance in this process. I have considerable reservations that the provided RoB assessments are accurate. For instance, primary prevention non-RCTS are retrospective, involving participants under completely different within study treatment regimens (duration, dosage, etc). I repeat within study differences (that means that participants were involved in a study irrespective of the exact intervention). In some cases, these elements (especially treatment duration) are not even specified. Moreover, the population analysed is not specified at all (population-based settings, outpatients seeking help, institutionalised individuals, etc). How come these methodological features were assessed as of low RoB. This is unacceptable. There must be similarity of exposures, otherwise there is a high risk of bias. The same applies to the secondary and tertiary prevention studies. In any case, please revise the whole RoB assessment plan (the tool you used might be inappropriate) and remove the expert-based evaluation of the strength of evidence.
3.4. Primary prevention: How come only 4 of the 5 studies contributed data on the risk of COVID-19 (you have mentioned that all 5 studies are relevant).
3.5. Secondary prevention: ‘‘Only one study, a retrospective cohort, reported our primary outcome’’. Same as above… The same comment applies to the part 3.6. Tertiary prevention (11 out of 13 studies contributed data to the main outcome). Please explain why were the remaining studies included if no relevant data were contributed.
- Discussion: The first 3 paragraphs are devoted to the recapitalization of your findings. Please condense your findings in a single paragraph and then proceed to discuss them. Moreover, please discuss (and provide in the results section) any study findings that were not meta-analysed. Your article is labelled as a systematic review and meta-analysis.
Finally, please attend to any inconsistencies, e.g., study [33] is described as both prospective (line195) and retrospective (line 206) in the text and as retrospective in the tables. Also, please remove vague comments such as ‘‘Three of five studies[15,29,32] adjusted their analysis for potential con- 239 founding variables such as age, gender and BMI.’’. These elements are important in the assessment of the quality of evidence and should be systematically considered in the context of the systematic review (meta-analytic approaches do not incorporate these features).
After applying the recommended changes, please revise your abstract. The number of studies meta-analysed and the findings of studies systematically assessed but not meta-analysed are not clear. I would advise you to consider systematically reviewing the evidence when you are not certain if it is appropriate to meta-analyse it.
Reviewer 2 Report
The manuscript titled "Effects of vitamin D supplementation on COVID-19 related 2 outcomes: a systematic review and meta-analysis", is an interesting systemic review shedding light on the impact of Vitamin D supplementation in different COVID-19 outcomes. The field is very fascinating, the paper is well written. The results are clearly reported and well discussed.
Author Response
Thank you very much for your close attention to our submitted manuscript.
Reviewer 3 Report
- Line 94: explain ED
- Line 168: Most (not mos)
- Line 293: RR = 0,46 (not 0,44)
Author Response
Thank you very much for your close attention to our submitted manuscript. Following is our response to your comments.
1. Line 94: explain ED
Action or explanation: Thank you for bringing this up to our attention. We have defined this abbreviation as requested.
2. Line 168: Most (not mos)
Action or explanation: Thank you for bringing this up to our attention. This error has been corrected.
3. Line 293: RR = 0,46 (not 0,44)
Action or explanation: Thank you for bringing this up to our attention. This error has been corrected.
Round 2
Reviewer 1 Report
Thank you for considering my suggestions. I hope you all the best!